# Detection of Serum-Specific IgE by Fluoro-Enzyme Immunoassay for Diagnosing Type I Hypersensitivity Reactions to Penicillins

**DOI:** 10.3390/ijms23136992

**Published:** 2022-06-23

**Authors:** Adriana Ariza, Cristobalina Mayorga, Gádor Bogas, Francesco Gaeta, María Salas, Rocco L. Valluzzi, Marina Labella, Natalia Pérez-Sánchez, Cristiano Caruso, Ana Molina, Tahia D. Fernández, María José Torres, Antonino Romano

**Affiliations:** 1Allergy Research Group, Instituto de Investigación Biomédica de Málaga-IBIMA, 29009 Malaga, Spain; adriana.ariza@ibima.eu (A.A.); lina.mayorga@ibima.eu (C.M.); gabhdor@hotmail.com (G.B.); mariasalascassinello@hotmail.com (M.S.); labellaalvarezmarina@gmail.com (M.L.); natbel.ps@gmail.com (N.P.-S.); anamolina22@hotmail.com (A.M.); tahiadfd@uma.es (T.D.F.); 2Allergy Unit, Hospital Regional Universitario de Málaga, 29009 Malaga, Spain; 3Andalusian Center for Nanomedicine and Biotechnology-BIONAND, 29590 Malaga, Spain; 4Allergy Unit, Columbus Hospital, Fondazione Policlinico Universitario Agostino Gemelli IRCCS, 00168 Rome, Italy; francesco.gaeta@policlinicogemelli.it; 5Department of Pediatrics, Division of Allergy, Pediatric Hospital Bambino Gesù, 00165 Rome, Italy; rvalluzzi@gmail.com; 6UOSD DH Gastroenterologia, Fondazione Policlinico Universitario Agostino Gemelli IRCCS, 00168 Rome, Italy; carusocristiano1@gmail.com; 7Departamento de Biología Celular, Genética y Fisiología, Universidad de Málaga, 29010 Malaga, Spain; 8Departamento de Medicina, Universidad de Málaga, 29010 Malaga, Spain; 9Oasi Research Institute-IRCCS, 94018 Troina, Italy; aromano.allergy@gmail.com

**Keywords:** allergy diagnosis, drug allergy, IgE, immunologic tests

## Abstract

Diagnosis of type I hypersensitivity reactions (IgE-mediated reactions) to penicillins is based on clinical history, skin tests (STs), and drug provocation tests (DPTs). Among in vitro complementary tests, the fluoro-enzyme immunoassay (FEIA) ImmunoCAP^®^ (Thermo-Fisher, Waltham, MA, USA) is the most widely used commercial method for detecting drug-specific IgE (sIgE). In this study, we aimed to analyze the utility of ImmunoCAP^®^ for detecting sIgE to penicillin G (PG) and amoxicillin (AX) in patients with confirmed penicillin allergy. The study includes 139 and 250 patients evaluated in Spain and Italy, respectively. All had experienced type I hypersensitivity reactions to penicillins confirmed by positive STs. Additionally, selective or cross-reactive reactions were confirmed by DPTs in a subgroup of patients for further analysis. Positive ImmunoCAP^®^ results were 39.6% for PG and/or AX in Spanish subjects and 52.4% in Italian subjects. When only PG or AX sIgE where analyzed, the percentages were 15.1% and 30.4%, respectively, in Spanish patients; and 38.9% and 46% in Italian ones. The analysis of positive STs showed a statistically significant higher percentage of positive STs to PG determinants in Italian patients. False-positive results to PG (16%) were detected in selective AX patients with confirmed PG tolerance. Low and variable sensitivity values observed in a well-defined population with confirmed allergy diagnosis, as well as false-positive results to PG, suggest that ImmunoCAP^®^ is a diagnostic tool with relevant limitations in the evaluation of subjects with type I hypersensitivity reactions to penicillins.

## 1. Introduction

Allergic reactions to beta-lactam (BL) antibiotics are a major health problem worldwide; these drugs are the most frequent cause of adverse drug reactions mediated by specific immunologic mechanisms [1,2]. The prevalence and incidence of these reactions change over time and across countries, possibly related to different patterns of prescription [2,3]. Nowadays, amoxicillin (AX), which is very often prescribed in combination with clavulanic acid (CLV) [2], is the most common elicitor of reactions in both children [4] and adults [2].

Based on the time interval between drug exposure and onset of symptoms, these reactions are classified as immediate (<1–6 h after the last drug exposure) or non-immediate (at any time from 1 h after the initial drug exposure) [5,6]. However, there is overlap in reactions appearing from 1 to 6 h, which was originally defined by Levine [5] as “accelerated reactions”. Immediate reactions are mostly mediated by specific IgE (sIgE) (type I hypersensitivity reactions [7]) and they commonly appear as isolated clinical manifestations, such as urticaria and angioedema, or as systemic reactions, such as anaphylaxis or anaphylactic shock [5,6,8,9]. The diagnosis of these reactions is complex and dynamic, especially due to the increased number of available chemical structures [10]. It is based on clinical history (CH), skin tests (STs), and drug provocation tests (DPTs), which are not risk-free procedures. The serum sIgE assay and the basophil activation test are often combined with STs as complementary methods to reduce the use of DPTs [11,12,13,14,15,16].

The detection of serum drug sIgE by the ImmunoCAP^®^ (Thermo-Fisher, Waltham, MA, USA), a standardized fluoro-enzyme immunoassay (FEIA), is the most widely used commercial method [11,12]. However, despite its great usefulness to detect allergen sIgE, in the diagnosis of allergic reactions to BLs, its sensitivity is variable and not optimal (0–50%) [17,18,19,20] and it could depend on the severity of the clinical symptoms [18]. Moreover, it is limited only to benzylpenicillin (penicillin G [PG]), penicillin V (PV), AX, ampicillin, and cefaclor [11,12]. In addition to sensitivity limitations, false-positive results with penicillin have been reported [21,22]. In order to improve sensitivity, the cut-off limit of ImmunoCAP^®^ has been lowered from 0.35 to 0.1 kUA/L. However, comparison of accuracy of the test with both cut-offs showed that 0.1 kUA/L, although increases sensitivity, it also reduces the specificity, particularly in subjects with total IgE (tIgE) > 200 kU/L [23]. A later study demonstrated that the application of sIgE/tIgE ratio (cut-off limit ≥ 0.002) significantly increases the positive likelihood ratio, the specificity, and the identification of true reactive patients, even among subjects with high levels of tIgE (tIgE > 200 kU/L) [24]. We have assessed the current diagnostic usefulness of ImmunoCAP^®^ for PG and AX for evaluating patients with type I hypersensitivity reactions to these penicillins. For that, we have analyzed its sensitivity in two well-characterized populations with different patterns of BL prescription (Spain and Italy).

## 2. Results

### 2.1. Patient Clinical Characteristics

The Spanish study group included 139 patients (72 females) with confirmed type I hypersensitivity reactions to penicillins (PG and/or AX) based on CH and ST. The percentage of positive STs to penicilloyl-poly-L-lysine (PPL)/benzylpenicilloyl-octa-L-lysine (BP-OL) and/or minor determinant mixture (MDM)/minor determinant (MD) was 13.7% and to AX 92.8%. The median age of the group was 45 years (interquartile range [IR]: 34–55) and the median time interval between the reaction and the allergy work-up was 180 days (IR: 90–330). The severity of the reaction was graded as grade 1 (mild), grade 2 (moderate), or grade 3 (severe); grade 2 was the most frequent (48.2%). The most commonly involved drug was AX (96.4%), administered as AX or as the combination AX-CLV (Table 1).

The Italian study group was composed of 250 patients (166 females) with confirmed type I hypersensitivity reactions to penicillins. The percentage of positive STs to PPL/BP-OL and/or MDM/MD was 46.8% and 96.4% to AX. The median age was 49 years (IR: 36.75–61) and the median time interval between the reaction and the allergy work-up was 60 days (IR: 30–180). The most frequent grade of reaction severity was grade 2 (50.4%) and the drug most commonly involved was AX (95.6%) (Table 1).

Comparative analysis between populations shows differences, with a significantly higher percentage of positive STs to PPL/BP-OL and/or MDM/MD in Italian patients. Moreover, a significantly shorter interval time between the reaction and the allergy work-up and a slightly older population was observed in the Italian group (Table 1).

When analyzing the percentage of patients with positive or negative ImmunoCAP^®^ to PG and/or AX according to the most recently recommended cut-off (≥0.1 kUA/L) [23], no significant differences regarding the clinical characteristics analyzed were found among Spanish patients. In Italian patients, however, the percentage of positive ImmunoCAP^®^ was significantly higher in those with positive STs to PPL/BP-OL and/or MDM/MD (Table 2).

### 2.2. Sensitivity and Specificity Values According to Cut-Off Criteria

The sensitivity and specificity values for the detection of PG and AX sIgE were compared for different cut-offs of the ImmunoCAP^®^, the most recently (≥0.1 kUA/L), and the previously (≥0.35 kUA/L) recommended by manufacturer and studies, as well as the sIgE/tIgE ratio (≥0.002) (Table 3). A cut-off of ≥0.10 kUA/L increased the sensitivity values for the detection of PG and/or AX sIgE from 15.5 to 39.6% in Spanish patients and from 32.4 to 52.4% in Italian patients compared with ≥0.35 kUA/L cut-off. However, the specificity of the test decreased from 93.8 to 68.8%. Independent analysis of the sensitivity for PG sIgE and AX sIgE, the values increased in the Spanish group from 6.5 to 15.1% and from 14.4 to 38.9%, respectively, and in the Italian group from 19.2 to 30.4% for PG and from 26.0 to 46.0% for AX; specificity decreased from 96.9 to 81.3% for PG and from 93.8 to 75.0% for AX. Due to the decrease in specificity when the ≥0.1 kUA/L cut-off was applied, and the increased specificity if subjects with tIgE ≥ 200 kU/L were excluded from the analysis (data not shown), an sIgE/tIgE ratio cut-off limit ≥ 0.002 was analyzed. However, we observed no differences with the percentage of positive ImmunoCAP^®^ obtained for ≥0.35 kUA/L cut-off, with a lower specificity value. According to these results, a cut-off limit of ≥0.1 kUA/L was applied in further analysis.

Comparing both study groups, the percentage of positive sIgE ImmunoCAP^®^ to PG was significantly higher in Italian patients (Table 3). These significant differences were not observed if patients from both study groups and with negative STs to PPL/BP-OL and MDM/MD and positive STs to AX were only included in the analysis.

### 2.3. Sensitivity of PG and AX sIgE Detection over Time

Changes in ImmunoCAP^®^ sensitivity along years were analyzed based on ImmunoCAP results obtained in consecutive five-year periods (2000–2004; 2005–2009; 2010–2014; 2015–2019) where each determination corresponds to a different patient. The percentage of positive results to PG and/or AX ranged from 31.4 to 52.8% in Spanish patients and from 47.4 to 55.5% in Italian ones. When only PG sIgE was analyzed, the percentage of positive results dropped to 9.4–19.4% in the Spanish group and to 25.6–40.9% in the Italian group. Finally, in the case of AX sIgE, values ranged between 31.3–52.8% and 40.9–50.8% for the Spanish and Italian population, respectively. In no case were statistically significant differences observed in the percentage of positivity between time periods analyzed for the same population. However, values were always higher in the Italian group compared with the Spanish group, with differences only significantly higher for PG sIgE in 2005–2009 (Figure 1). Levels of PG and AX sIgE in positive ImmunoCAP^®^ cases only showed significant differences between Spain and Italy for AX in 2010–2014 (Figure 2). The analysis of positive STs to PG determinants (PPL/BP-OL and/or MDM/MD) and to AX showed a statistically significant higher percentage of positive STs to PG determinants in the Italian group in all the analyzed time periods (Figure 3A). The time interval between the reaction and the allergy work-up was significantly lower in Italian patients (Figure 3B).

### 2.4. Sensitivity of PG and AX sIgE and Specificity of PG sIgE in Selective/Cross-Reactive Patients

Of the 139 Spanish patients, 53 were classified into AX selective or PG_AX cross-reactive as follows: AX selective when they presented negative STs to PG determinants and confirmed tolerance to PG by a negative DPT to PG or PV, and PG_AX cross reactive, when they displayed positive STs and/or DPTs to both PG (and/or PG determinants) and AX; 25 patients were AX selective reactors and 28 patients were PG_AX cross-reactors. The percentage of positive sIgE ImmunoCAP^®^ to AX was 48.0% and 39.3% for selective and cross-reactive patients, respectively. In the case of PG sIgE, the percentage of positive cases was 16.0% for AX selective patients (false-positive results) and 17.7% PG_AX cross-reactor patients (Figure 4).

## 3. Discussion

The diagnosis of allergic reactions to penicillins includes the CH, STs, DPTs, and in vitro tests, with the latter as relevant diagnostic tools for safety reasons [11,18]. The most frequently used in vitro test is the commercialized available ImmunoCAP^®^ System [17]. However, one of the main limitations of this test is the low sensitivity in the case of BL sIgE, with values in several studies [17,18,19,20,23,24,25] reported to range between 0–50%, and specificity values from 85.7 to 100% depending on the study and the severity of the reaction. In the present study, we have analyzed the value of the ImmunoCAP^®^ as diagnostic tool in two different groups with differences in the BL prescription, Spanish (N = 139) and Italian (N = 250); patients were diagnosed by positive STs and evaluated from 2000 to 2019 and from 2005 to 2019, respectively. This was used as the selection criteria since although to confirm the clinical value of an in vitro test the ideal approach would be a blind comparison to a reference standard like DPT, this cannot be performed in all patients for ethical reasons. Thus, in most studies, a convincing CH and positive STs are considered the ‘‘reference test’’ [11,26]. Both study groups were significantly different in terms of a higher percentage of positive STs to PPL/BP-OL and/or MDM/MD in the Italian group, which was related to a different sensitization pattern, similar to the pattern observed in the Spanish population more than twenty years ago [27,28], and with a higher percentage of positive sIgE ImmunoCAP^®^ to PG.

When analyzing the percentage of patients with positive or negative ImmunoCAP^®^ to PG and/or AX, no significant differences in the related clinical characteristics were found among Spanish patients; however, in Italian patients, the percentage of positive ImmunoCAP^®^ was significantly higher in those with positive STs to PPL/BP-OL and/or MDM/MD. The obtained sensitivity value was 39.6% and 52.4% in Spanish and Italian patients, respectively, and specificity was 68.8% when both haptens, PG and AX, were analyzed. According to these results, only 39.6% (Spain) and 52.4% (Italy) of subjects with PG and/or AX allergy confirmed by STs could have been diagnosed by ImmunoCAP^®^, values that might be lower in patients diagnosed by DPTs and when ImmunoCAP^®^ is widely applied to the diagnosis of patients with suspected allergy to penicillins, which limits the usefulness of ImmunoCAP^®^ as complementary diagnostic tool to in vivo tests. The significantly higher sensitivity values obtained in the Italian group would be related to the significantly higher percentage of positive STs to PG determinants; indeed, if these patients are excluded from sensitivity analysis, no significant differences are detected between the Spanish and Italian groups. Moreover, a significantly lower time interval between the reaction and the allergy work-up in the Italian group could have also contributed to the higher percentage of positive ImmunoCAP^®^ results, as previous studies have shown that drug sIgE decreases with time [29]; hence, the European Network of Drug Allergy (ENDA) recommends that the assay should not be performed after more than 3 years after the reactions [11], as considered in the inclusion criteria for this study.

The sensitivity and specificity values obtained for the most recently considered cut-off limit (≥0.1 kUA/L) were compared with the data obtained if the previous cut-off limit (≥0.35 kUA/L) was applied. Previous studies have reported a significant increase in sensitivity using the cut-off of ≥0.10 kUA/L compared with ≥0.35 kUA/L, but with a strong parallel impairment concerning specificity [23]. Indeed, PG and/or AX ImmunoCAP^®^ sensitivity dropped to 15.8% and 32.4% in Spanish and Italian patients, respectively, when a cut-off limit ≥0.35 kUA/L was applied, with specificity increasing up to 93.8%. Focusing on specificity, previous studies showed that tIgE values influence the diagnostic performance of serum sIgE assays, with a diagnostic odds ratio breaking down for tIgE values above 200 or 500 kU/L [23]. Our results also show that specificity increased up to 83.3% (cut-off ≥0.1 kUA/L) and 100% (cut-off ≥0.35 kUA/L) when only tolerant subjects with tIgE values <200 kU/L were included as negative controls. The correlation of penicillin sIgE and tIgE detected in tolerant patients suggests that the source for false-positive results is that the system wrongly measures a fraction of tIgE as sIgE, as suggested in a recent study [24]. In that sense, the study by Vultaggio evaluated the usefulness of serum BL sIgE/tIgE ratio for improving the performance characteristics of the assay. The study established a sIgE/tIgE ratio of ≥0.002 as the cut-off limit for 95% specificity and 43.3% sensitivity [24], suggesting that the application of sIgE/tIgE ratio is better than conventional positivity. However, according to our results, the sIgE/tIgE ratio increases the specificity but decreases the sensitivity until values obtained for cut-off ≥0.35 kUA/L.

The percentage of positive ImmunoCAP^®^ results was analyzed over time comparing five-year periods (2000–2004; 2005–2009; 2010–2014; and 2015–2019) and showed no statistically significant differences in the same study group. However, values were always higher in the Italian group compared with the Spanish group, although differences were only significantly higher for PG sIgE for 2005–2009. The higher percentage of positive results in Italian patients might be associated with a statistically significant higher percentage of positive STs to PG determinants for all the time periods analyzed, as well as a significantly shorter time between the reaction and the allergy work-up, as mentioned above. In the analysis of a subset of Spanish patients, PG_AX cross-reactive patients showed 17.7% of positive PG sIgE, similar to previous reports [19]. However, it is important to highlight that 16.0% of positive results for PG sIgE were obtained in AX selective patients who tolerated PG; therefore, these results should be considered as false-positive results. As previously reported [22], they were likely due to not clinically relevant sIgE to phenylethylamine, a structure with a benzyl group that can be present on the ImmunoCAP^®^ and that shares allergenic epitopes with PG and PV. In summary, in this study, we have found low and variable ImmunoCAP^®^ sensitivity values in two well-defined study groups with PG and/or AX allergy confirmed by STs, as well as false-positive results to PG in a subgroup of patients with PG tolerance confirmed by DPTs. All these observations suggest that the ImmunoCAP^®^ is a diagnostic tool with relevant limitations in the evaluation of subjects with type I hypersensitivity reactions to penicillins. Further prospective studies in non-selected subjects with suspicion of type I hypersensitivity to penicillins should be performed to established the real predictive value of this in vitro method.

## 4. Materials and Methods

### 4.1. Study Design and Participants

Retrospective study conducted according to STARD guidelines (https://www.equator-network.org/reporting-guidelines/stard/, accessed on 9 June 2022) [30]. The study includes consecutive patients evaluated in the Allergy Unit of the Hospital Regional Universitario de Málaga (Spain) (period 2000–2019) and in the Allergy Units of the Columbus Hospital, Rome (Italy) and Oasi Research Institute-IRCCS, Troina (Italy) (period 2005–2019). The inclusion criteria were patients with a confirmed diagnosis of type I hypersensitivity reaction to penicillins based on the CH and positive STs and with and interval time between the last penicillin reaction and the allergy work-up of no longer than 3 years, as recommended by ENDA [11]. Additionally, patients with no contra-indication to perform DPT, were confirmed as cross-reactors to penicillins or selective reactors to AX for further analysis. Severity of the reactions was graded as: grade 1 (mild: skin and subcutaneous tissues); grade 2 (moderate: features suggesting respiratory, cardiovascular, or gastrointestinal involvement); or grade 3 (severe: hypoxia, hypotension, or neurologic compromise). Thirty subjects with confirmed tolerance to penicillins were included as negative controls. The study was conducted according to the Declaration of Helsinki principles and the protocol was approved by the respective institutional review boards. Prior to the allergy work-up, all subjects received information about the possible risks of STs and DPTs, and written informed consent was obtained from each patient or the representatives of those under 18 years of age.

### 4.2. In Vivo Diagnostic Methods for Patient Evaluation

#### 4.2.1. Skin Testing

Skin prick tests (SPTs) and, if negative, intradermal tests (IDTs) were performed as recommended in the European Academy of Allergy and Clinical Immunology (EAACI) guidelines [11,12,15,31] using solutions prepared daily. The maximum concentration of penicillin derivatives used was: PPL 1.07 × 10^−2^ M; MDM (benzylpenicillin, benzylpenicilloate, and benzylpenilloate) 1.5 M; and AX 20 mg/mL (5 × 10^−2^ M) (all from Diater laboratories, Madrid, Spain). Since May 2011, the composition of penicillin allergenic determinants changed and included the major determinant BP-OL 0.04 mg/mL (8.64 × 10^−5^ M benzylpenicilloyl moiety), and the MD 0.5 mg/mL (1.5 × 10^−3^ M sodium benzylpenilloate). In the SPT reading, a wheal larger than 3 mm surrounded by erythema, with a negative response to the control saline, was considered positive. For IDTs, the wheal area was marked at the beginning and 20 min after testing. An increase in the wheal diameter greater than 3 mm surrounded by erythema was considered positive [32].

#### 4.2.2. Drug Provocation Test

In patients with ST positive to AX and negative to PPL/BP-OL and MDM/MD, a graded DPT with PG and PV was proposed. In consenting patients, DPTs were carried out in a single-blind procedure according to the EAACI guidelines [15,31]. The cumulative doses administered were PG 106 IU (Normon Laboratories, Madrid, Spain), intramuscularly, or PV 400 mg (ERN, Barcelona, Spain), orally. Those reacting to PG or PV were considered as cross-reactors and those with good tolerance as selective reactors to AX.

### 4.3. In Vitro Determination of Serum sIgE

The FEIA ImmunoCAP^®^ (Thermo-Fisher) was used for detecting serum sIgE to penicilloyl G (c1) and amoxicilloyl (c6) in patients and controls following the manufacturer’s instructions. Results were obtained by direct comparison with standards run in parallel (detection limit <0.10 kUA/L) and values were analyzed and compared with two cut-offs (≥0.1 kUA/L and ≥0.35 kUA/L). Serum tIgE was also assayed by the ImmunoCAP^®^ according to the manufacturer’s instructions.

### 4.4. Statistical Analysis

Quantitative variables were expressed as mean and standard deviation (SD) or median and IR, and comparisons were carried out using the Mann–Whitney and Wilcoxon tests for median values of nonrelated and related samples, respectively. Comparisons between qualitative variables were analyzed by Chi-square test or Fisher’s exact test, and *p* values of ≤0.05 were considered statistically significant and presented as * *p* ≤ 0.05, ** *p* < 0.01, *** *p* < 0.001, or **** *p* < 0.0001. Bonferroni correction was carried out for multiple testing. The statistical analysis was performed using the GraphPad Prism program, version 7.05.

## Figures and Tables

**Figure 1 ijms-23-06992-f001:**
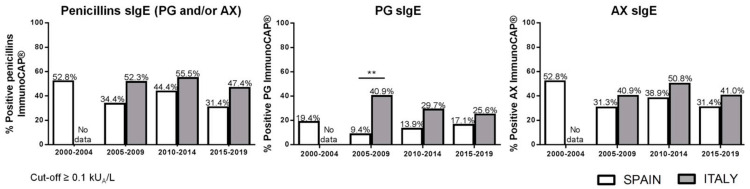
Percentage of patients with positive specific IgE (sIgE) to penicillin G (PG) and/or amoxicillin (AX) in Spanish and Italian groups. Significant differences if *p* ≤ 0.05: ** *p* < 0.01.

**Figure 2 ijms-23-06992-f002:**
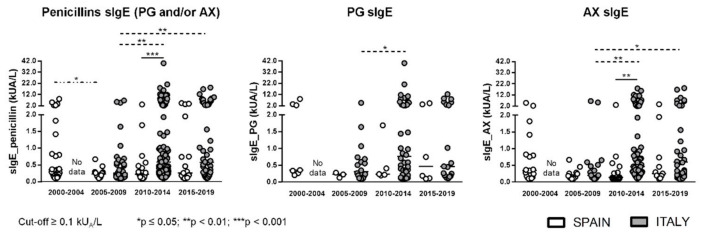
sIgE levels (kUA/L) to PG and/or AX, to PG, or to AX in both Spanish and Italian patients with positive ImmunoCAP^®^. Significant differences if *p* ≤ 0.05.

**Figure 3 ijms-23-06992-f003:**
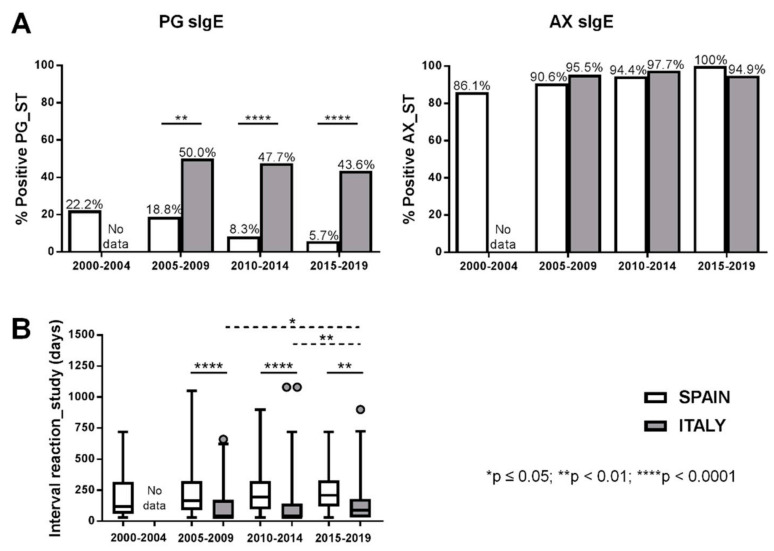
(**A**) Time interval between the reaction and the allergy work-up. (**B**) Percentage of patients with positive skin tests PG or to AX over time. Significant differences if *p* ≤ 0.05.

**Figure 4 ijms-23-06992-f004:**
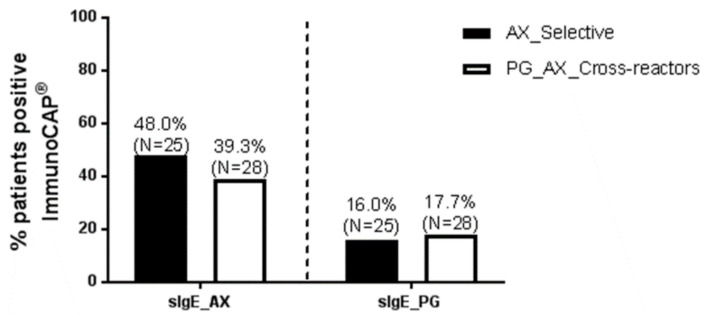
Percentage of positive sIgE to PG or to AX in a subgroup of Spanish patients confirmed as AX selective or PG_AX cross-reactors by drug provocation tests.

**Table 1 ijms-23-06992-t001:** Clinical characteristics, reaction severity, drugs involved, and skin test results for patients included in the study, and statistical comparisons of the Spanish and Italian study groups.

		Spain (N = 139)	Italy (N = 250)	*p* ^1^
Median (IR) age, years		45 (34–55)	49 (36.75–61)	0.0088
Gender, % female		72/139 (51.8%)	166/250 (66.4%)	0.0065
Reaction severity, N° (%) patients	Grade 1	35/139 (25.2%)	70/250 (28.0%)	ns
	Grade 2	67/139 (48.2%)	126/250 (50.4%)	ns
	Grade 3	37/139 (26.6%)	54/250 (21.6%)	ns
Drug causing allergic reaction, N° (%) patients	Penicillin	4/139 (2.9%)	0/250 (0.0%)	-
	AX/AX-CLV	134/139 (96.4%)	239/250 (95.6%)	ns
	AMP/BAC	1/139 (0.7%)	11/250 (4.4%)	-
Positive N° (%) skin tests	PPL/BP-OL, MDM/MD	19/139 (13.7%)	117/250 (46.8%)	<0.0001
	AX	129/139 (92.8%)	241/250 (96.4%)	ns
Median (IR) time interval reaction/study, days		180 (90–330)	60 (30–180)	<0.0001

^1^ Differences were considered statistically significant if *p* ≤ 0.05. AMP: ampicillin; AX: amoxicillin; BAC: bacampicillin; BP-OL: benzylpenicilloyl-octa-L-lysine; CLV: clavulanic acid; MD: minor determinant; MDM: minor determinant mixture; ns: not significant; PPL: penicilloyl-poly-L-lysine; IR: interquartile range.

**Table 2 ijms-23-06992-t002:** Clinical characteristics and allergy work-up by specific IgE ImmunoCAP^®^ results (cut-off ≥ 0.1 kUA/L) for Spanish and Italian study groups. Statistical comparison shown is between positive and negative ImmunoCAP^®^.

		Spain			Italy		
		Positive ImmunoCAP (PG and/or AX) (N = 55)	Negative ImmunoCAP (PG and AX) (N = 84)	*p* ^1^	Positive ImmunoCAP (PG and/or AX) (N = 131)	Negative ImmunoCAP (PG and AX) (N = 119)	*p* ^1^
Median (IR) age, years		46 (34–56)	45 (35–53)	ns	48 (35–57)	53 (38–65)	0.0205
Gender, % female		28/55 (50.9%)	44/84 (52.4%)	ns	87/131 (66.4%)	79/119 (66.4%)	ns
Reaction severity, N° (%) patients	Grade 1	10/55 (18.2%)	25/84 (29.8%)	ns	30/131 (22.9%)	40/119 (33.6%)	ns
	Grade 2	31/55 (56.4%)	36/84 (42.9%)	ns	70/131 (53.4%)	56/119 (47.1%)	ns
	Grade 3	14/55 (25.5%)	23/84 (27.4%)	ns	31/131 (23.7%)	23/119 (19.3%)	ns
Drug causing allergic reaction, N° (%) patients	Penicillin	0/55 (0.0%)	4/84 (4.76%)	-	0/131 (0.0%)	0/119 (0.0%)	-
	AX/AX-CLV	55/55 (100.0%)	79/84 (94.1%)	ns	126/131 (96.2%)	113/119 (95.0%)	ns
	AMP/BAC	0/55 (0.0%)	1/84 (1.2%)	-	5/131 (3.8%)	6/119 (5.0%)	ns
Positive N° (%) skin tests	PPL/BP-OL, MDM/MD	9/55 (16.4%)	10/84 (11.9%)	ns	72/131 (55.0%)	45/119 (37.8%)	0.0078
	AX	51/55 (92.7%)	78/84 (92.9%)	ns	123/131 (93.9%)	118/119 (99.2%)	ns
Median (IR) time interval reaction/study, days		150 (90–300)	180 (90–330)	ns	60 (30–120)	60 (30–180)	ns

^1^ Differences were considered statistically significant if *p* ≤ 0.05. PG: penicillin G.

**Table 3 ijms-23-06992-t003:** Sensitivity and specificity values for the detection of specific IgE to penicillin G and amoxicillin by ImmunoCAP^®^ comparing different cut-offs.

	Pencillin sIgE (PG and/or AX sIgE)				PG sIgE				AX sIgE			
Cut-Off	Sens (SP)	Sens (IT)	*p* ^1^	Spec	Sens (SP)	Sens (IT)	*p* ^1^	Spec	Sens (SP)	Sens (IT)	*p* ^1^	Spec
sIgE ≥ 0.1 kUA/L	39.6% (N = 139)	52.4% (N = 250)	0.0197	68.8% (N = 32)	15.1% (N = 139)	30.4% (N = 250)	0.0009	81.3% (N = 32)	38.9% (N = 139)	46.0% (N = 250)	ns	75.0% (N = 32)
sIgE ≥ 0.35 kUA/L	15.8% (N = 139)	32.4% (N = 250)	0.0003	93.8% (N = 32)	6.5% (N = 139)	19.2% (N = 250)	0.0005	96.9% (N = 32)	14.4% (N = 139)	26.0% (N = 250)	0.0101	93.8% (N = 32)
sIgE/tIgE ≥ 0.002	13.7% (N = 139)	35.7% (N = 244)	<0.0001	81.3% (N = 32)	8.6% (N = 139)	20.1% (N = 244)	0.0034	87.5% (N = 32)	12.2% (N = 139)	29.5% (N = 244)	0.0001	90.6% (N = 32)

^1^ Differences were considered statistically significant if *p* ≤ 0.05. Bonferroni correction alpha = 0.017. IT: Italy; sens: sensitivity; sIgE: specific IgE; SP: Spain; spec: specificity; tIgE: total Ig.

## Data Availability

All data are contained within the manuscript. Raw data are available on reasonable request from the corresponding author.

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
