# Peer review of "Detection of Serum-Specific IgE by Fluoro-Enzyme Immunoassay for Diagnosing Type I Hypersensitivity Reactions to Penicillins"

_ijms, 2022, doi:10.3390/ijms23136992_

Round 1

Reviewer 1 Report

The study appears to be a diagnostic accuracy study, so it needs to be rewritten in an appropriate format and reference to guidelines made, e.g.1  The aim being to determine the diagnostic accuracy of a new cut-off vs the previous gold standard. It might be clearer to refer to gold standard and new test, “recent” and “previous” is not so clear.

The approach is currently confused – was the aim to determine the diagnostic accuracy of a new test cut-off or to look at difference in reactivity in 2 different populations or to look at reactivity changes over time?  I would suggest choosing one research question and focussing on that. In the former, all the patients should be assessed together, with details of how patients were selected. This would have the advantage of assessing performance across the range of geographical areas, and provide an overview of diagnostic accuracy.

Methods

Aside from reformatting according to STARD:

To allow readers to understand the risk of bias in subject selection please state method of identification of subjects, study inclusion, and exclusion criteria.

Please provide a flow chart of patient identification methods, reasons for exclusion of any patients.

Please clarify how the recruitment processes and patient selection differed in the 2 centres; it seems unlikely that it was the same.

Sample size calculations are missing.

Statistical analysis

Was correction for multiple testing carried out?

Include definitions used for ‘clinical entities’ line 88 and who determined these diagnoses.

The study group should be patient diagnosed with clinical features of a type 1 hypersensitivity reaction with positive skin test

Minor comments

Restrict decimal places to 1, study sample size is not large enough to warrant 2 decimal places.

Table 2 please rephrase “drug involved..” to ?”drug causing index allergic reaction”

1.         Bossuyt PM, Reitsma JB, Bruns DE, et al. STARD 2015: an updated list of essential items for reporting diagnostic accuracy studies. Bmj 2015; 351: h5527.

Author Response

Point 1: The study appears to be a diagnostic accuracy study, so it needs to be rewritten
in an appropriate format and reference to guidelines made, e.g.1 The aim being to
determine the diagnostic accuracy of a new cut-off vs the previous gold standard. It
might be clearer to refer to gold standard and new test, “recent” and “previous” is not so
clear. The approach is currently confused – was the aim to determine the diagnostic
accuracy of a new test cut-off or to look at difference in reactivity in 2 different
populations or to look at reactivity changes over time? I would suggest choosing one
research question and focusing on that. In the former, all the patients should be assessed
together, with details of how patients were selected. This would have the advantage of
assessing performance across the range of geographical areas, and provide an overview
of diagnostic accuracy.

Response 1: We thank the referee for the comments and for the opportunity to clarify
this point. The aim of this study was not to determine the diagnostic accuracy of a new
cut-off vs the previous gold standard, as this new cut-off has been analyzed by previous
studies and manufacturer instructions. The aim of this study was to analyze the
usefulness of this method in the diagnosis of immediate allergic reactions. To this aim,
we have analyzed the sensitivity of the test in two well-defined groups of patients
(Spanish and Italian) with immediate reactions confirmed by positive skin test. We
analyzed both new cut-off and the previous gold standard, in order to confirm that the
sensitivity with the new cut-off was higher in our study groups. Moreover, we also
analyzed the sensitivity of the test in consecutive five-year periods (with different
patients per time period) to observe if sensitivity changes along years were detected, and
in that case, if these changes could be associated to changes in the sensitization pattern
of the population. However, no significant sensitivity changes have been observed, as
shown in the manuscript. Modifications have been included in the text to clarify these
comments and suggestions.

Point 2: Methods

Aside from reformatting according to STARD:

To allow readers to understand the risk of bias in subject selection please state method
of identification of subjects, study inclusion, and exclusion criteria.

Please provide a flow chart of patient identification methods, reasons for exclusion of
any patients.

Please clarify how the recruitment processes and patient selection differed in the 2
centres; it seems unlikely that it was the same.

Response 2: We thank the referee for the comments. Changes according to STARD
have been included in Methods. Recruitment processes and patient selection were the
same in the centres by following the European Academy of Allergy and Clinical
Immunology (EAACI) guidelines (Romano et al., Allergy.
2020;75(6):1300-1315).
Point 3: Sample size calculations are missing.

Response 3: We thank the referee for the opportunity to clarify this point. This is a
retrospective study
which includes the entire population evaluated in the study centres

according to the inclusion criteria of the study: immediate allergic patients to
penicilllins confirmed by positive ST and an interval time between the last penicillin
reaction and the allergy work-up no longer than 3 years. Nevertheless,
the estimated
sample size was 346 for a diagnostic sensitivity of penicillin ImmunoCAP > 30%,
a
non-allergic/allergic ratio of 0.07, a confidence level of 95% and a precision of 5%.

Point 4: Statistical analysis. Was correction for multiple testing carried out?

Response 4: We thank the referee for the comment. Bonferroni correction alpha has
been included in tables 2 and 3.

Point 5: Include definitions used for ‘clinical entities’ line 88 and who determined these
diagnoses.

Response 5: We thank the referee for the comment. Clinical entities have been changed
by grade severity of immediate reactions (Brown SG. J Allergy Clin Immunol
2004;114(2):371-6). Definitions for grade 1 (mild), grade 2 (moderate), and grade 3
(severe) have been included in methods section.

Point 6: The study group should be patient diagnosed with clinical features of a type 1
hypersensitivity reaction with positive skin test.

Response 6: We agree with the referee, immediate reaction has been replaced by type I
hypersensitivity reactions.

Minor comments

Point 7: Restrict decimal places to 1, study sample size is not large enough to warrant 2
decimal places.

Response 7: Changes have been made in text and tables to restrict decimal places to 1.

Comment 8: Table 2 please rephrase “drug involved..” to ?”drug causing index allergic
reaction”.

Response 8: “Drug involved...” has been rephrased in tables 1 and 2.

1. Bossuyt PM, Reitsma JB, Bruns DE, et al.
STARD 2015: an updated list of
essential items for reporting diagnostic accuracy studies. Bmj 2015; 351: h5527.

Reviewer 2 Report

The article ˝Detection of serum specific IgE by fluoro-enzyme immunoassay for diagnosing immediate allergic reactions to penicillins˝ provides the foundation for diagnosing allergic reactions, which is greatly meaningful. However, there are some problems that must be addressed or answered before publication as follow: 1) there are inaccurate citations in the manuscript, 2) clinical symptoms must be refined accurately, 3) The study time is 2000-2019, in which patients may die or the ImmunoCAP® may be updated. I don't find these considerations in the text. So the data (Fig. 1~Fig. 3) may be unconvincing or unreliable, which must be answered. Additionally, what does it mean to evaluate ˝PG and AX sIgE detection over time˝ (section 2.3 and section 2.4)? Sections 2.3 and 2.4 have no meaning for diagnosing allergic reactions. For detection methods, the accuracy, specificity and detection limit are the most concerned, which should be mainly considered in the manuscript. 4) You should assess the cross-reaction in detail.  

Some specific comments are as follow:

Line 43: What’s “specific immunologic mechanisms”? Please explain it.

Line 45-46: The sentence is uncomfortable. Please re-write it. The author wants to demonstrate that “amoxicillin (AX) and clavulanic acid (CLV) are prescription drugs, in which AX is the most common elicitor in both children and adults nowadays”?

Line 48-50: Please cite references correctly and reasonably.

For the reference 5 (Immunologic mechanisms of penicillin allergy. A haptenic model system for the study of allergic diseases of man), the drug reaction is divided into immediate reactions (occur within two to twenty minutes), accelerated reactions (begin between two and forty-eight hours), and late reactions (begin more than three days).

For the reference 6 (International Consensus on drug allergy), the drug reaction is divided into immediate reactions (occur within 1–6 h after the last drug administration) and nonimmediate reactions (occur at any time as from 1 h after the initial drug administration), which is consistent with your classification.

Line 52: The reference 8 states “We propose that the term ‘‘immediate’’ for likely IgE mediated or pseudoallergic reactions presenting with urticaria, angioedema, bronchospasm, or anaphylaxis should encompass also these manifestations occurring at up to several hours (eg, <_6 hours)”. Therefore, the statement “Immediate reactions are mostly mediated by specific IgE (sIgE)” is inaccurate.

Line 53-54: From the reference 9, anaphylaxis is a severe, life-threatening generalized or systemic hypersensitivity reaction, including anaphylactic shock. Therefore, the sentence is inaccurate.

Line 65: The author state “it is limited only to benzylpenicillin (penicillin G [PG]), penicillin V (PV), AX, ampicillin, and cefaclor”. Why the author only assessed the current diagnostic usefulness of ImmunoCAP® for PG and AX rather than PG, PV, AX, ampicillin, and cefaclor.

Line 88-89: anaphylaxis is a severe, life-threatening generalized or systemic hypersensitivity reaction, including anaphylactic shock and death. Therefore, the clinical symptoms should be refined.

Line 202: depending on the study? Method?

Therefore, I recommend a major revision based on the above comments.

Author Response

The article ̋Detection of serum specific IgE by fluoro-enzyme immunoassay for
diagnosing immediate allergic reactions to penicillins ̋ provides the foundation for
diagnosing allergic reactions, which is greatly meaningful. However, there are some
problems that must be addressed or answered before publication as follow:

Point 1: there are inaccurate citations in the manuscript.

Response 1: We agree with the referee. Inaccurate citations have been modified in the
manuscript.

Point 2: clinical symptoms must be refined accurately.

Response 2: We thank the referee for the comment. Clinical entities have been
redefined according to grading system for generalized hypersensitivity reactions by
Brown SG (J Allergy Clin Immunol 2004;114(2):371-6. Definitions and reference have
been included in method sections and required changes have been included in text and
tables.

Point 3: The study time is 2000-2019, in which patients may die or the ImmunoCAP®
may be updated. I don't find these considerations in the text. So the data (Fig. 1~Fig. 3)
may be unconvincing or unreliable, which must be answered.

Response 3: We thank the referee for the opportunity to clarify this point. The study
time was not a longitudinal study to analyze changes in the same patients over time. In
this study, we analyzed the sensitivity of the test based on ImmunoCAP results obtained
in consecutive five-years periods (2000-2004; 2005-2009; 2010-2014; 2015-2019)
where each determination corresponds to a different patient. For that, possible death of
some patients should not be considered in this study. Regarding ImmunoCAP update,
cut-off was lowered by manufacturer ́s instructions (> 0.1 kUA/L), and this aspect was
analyzed in the manuscript. The aim of the study time was to analyze if changes in
ImmunoCAP sensitivity along years were significant, and in that case, if these changes
could be associated to modifications in the sensitization pattern of the general
population. Modifications have been included in the text to clarify these comments.

Point 4: Additionally, what does it mean to evaluate ̋PG and AX sIgE detection over
time ̋ (section 2.3 and section 2.4)? Sections 2.3 and 2.4 have no meaning for
diagnosing allergic reactions. For detection methods, the accuracy, specificity and
detection limit are the most concerned, which should be mainly considered in the
manuscript.

Response 4: We thank the referee for the comments. As mentioned above, the aim of
the study time (section 2.3) was to analyze if changes in ImmunoCAP sensitivity along
years were significant, and in that case, if these changes could be associated to
modifications in the sensitization pattern of the general population. On the other hand,
we made a mistake in the title of the section 2.4, which has been modified in the

manuscript. Accuracy, specificity, and detection limit have been considered in the
manuscript.

Point 5: You should assess the cross-reaction in detail.

Response 5: We agree with referee. Unfortunately we have no possibility to perform a
more detailed cross-reaction study because of the number of patients in which DPT was
indicated to confirm selective or cross-reactive reactions compared with the whole study
group was low (N=53), and for ethical reason we were not able to increase the study
group.

Some specific comments are as follow:

Point 6: Line 43: What’s “specific immunologic mechanisms”? Please explain it.

Response 6: Reactions mediated by specific immunologic mechanisms are reactions
mediated by drug-specific antibodies, drug-specific T lymphocytes, IgG or IgM
(cytotoxic reactions), or immune-complex (Gell and Coombs, Clinical aspects in
immunology, Blackwell Scientific Publications, 1968).

Point 7: Line 45-46: The sentence is uncomfortable. Please re-write it. The author
wants to demonstrate that “amoxicillin (AX) and clavulanic acid (CLV) are prescription
drugs, in which AX is the most common elicitor in both children and adults nowadays”?

Response 7: We thank the referee for the comment. Sentence has been re-written.

Point 8: Line 48-50: Please cite references correctly and reasonably. For the reference 5
(Immunologic mechanisms of penicillin allergy. A haptenic model system for the study
of allergic diseases of man), the drug reaction is divided into immediate reactions (occur
within two to twenty minutes), accelerated reactions (begin between two and forty-eight
hours), and late reactions (begin more than three days).

Response 8: Reference has been changed.

Point 9: For the reference 6 (International Consensus on drug allergy), the drug reaction
is divided into immediate reactions (occur within 1–6 h after the last drug
administration) and nonimmediate reactions (occur at any time as from 1 h after the
initial drug administration), which is consistent with your classification.

Response 9: Reference has been changed.

Point 10: Line 52: The reference 8 states “We propose that the term ‘‘immediate’’ for
likely IgE mediated or pseudoallergic reactions presenting with urticaria, angioedema,
bronchospasm, or anaphylaxis should encompass also these manifestations occurring at
up to several hours (eg, <_6 hours)”. Therefore, the statement “Immediate reactions are
mostly mediated by specific IgE (sIgE)” is inaccurate.

Response 10: Reference has been changed.

Point 11: Line 53-54: From the reference 9, anaphylaxis is a severe, life-threatening
generalized or systemic hypersensitivity reaction, including anaphylactic shock.
Therefore, the sentence is inaccurate.

Response 11: Reference has been changed.

Point 12: Line 65: The author state “it is limited only to benzylpenicillin (penicillin G
[PG]), penicillin V (PV), AX, ampicillin, and cefaclor”. Why the author only assessed
the current diagnostic usefulness of ImmunoCAP® for PG and AX rather than PG, PV,
AX, ampicillin, and cefaclor.

Response 12: We thank the referee for the comment. PV, ampicillin, and cefaclor were
not included in the study because of the low number of ImmunoCAP determinations to
these drugs in the study populations, compared to PG and AX in both groups. For that
reason, unfortunately, it was not possible to carry out statistical studies for PV,
ampicillin and cefaclor.

Point 13: Line 88-89: anaphylaxis is a severe, life-threatening generalized or systemic
hypersensitivity reaction, including anaphylactic shock and death. Therefore, the
clinical symptoms should be refined.

Response 13: We thank the referee for the comment. Clinical entities have been
redefined according to grading system for generalized hypersensitivity reactions by
Brown SG (J Allergy Clin Immunol 2004;114(2):371-6). Definitions and reference have
been included in method sections and required changes have been included in text and
tables.

Point 14: Line 202: depending on the study? Method?

Response 14: We thank the referee for the opportunity to clarify this point. Values
ranged depending on the study design and the clinical characteristics of the patients
included in each study.

Round 2

Reviewer 2 Report

The author has responded well to my review comments.I recommend accepting in present form.